# Exogenous of Indole-3-Acetic Acid Application Alleviates Copper Toxicity in Spinach Seedlings by Enhancing Antioxidant Systems and Nitrogen Metabolism

**DOI:** 10.3390/toxics8010001

**Published:** 2019-12-24

**Authors:** Qin Gong, Zhaohua Li, Ling Wang, Tongwei Dai, Qun Kang, Duandan Niu

**Affiliations:** 1Faculty of Resources and Environmental Science, Hubei University, Wuhan 430062, China; 201701110700051@stu.hubu.edu.cn (Q.G.); wlk87@outlook.com (L.W.); 201711110811086@stu.hubu.edu.cn (T.D.); Kangqun@hubu.edu.cn (Q.K.); 201811110811197@stu.hubu.edu.cn (D.N.); 2Further Education Colleges, Xinjiang Vocation College of Agriculture, Changji 831100, China; 3Hubei Rural Safe Drinking Water Engineering Technology Research Center, Wuhan 430062, China

**Keywords:** indole-3-acetic acid, copper stress, antioxidant enzyme activity, nitrogen metabolism, spinach seedlings

## Abstract

Indole-3-acetic acid (IAA) is a potential mediator in the protection of plants from copper (Cu) toxicity and the enhancement of Cu tolerance. In this paper, spinach (*Spinacia oleracea* L.) seedlings were cultivated in soil containing 700 mg kg^−1^ Cu and the leaves of seedlings were sprayed with different concentrations of IAA. Exogenous IAA treatment reduced the malondialdehyde (MDA) concentrations in Cu-stressed seedlings and increased biomass, proline content, and the activities of antioxidant enzymes. Exogenous IAA treatment also increased the levels of nitrogen (N) assimilation compounds and the activities of N-metabolizing enzymes, but reduced NH_4_^+^ content. Notably, lower concentrations of IAA (10–40 mg L^−1^) increased the Cu concentrations in roots and reduced the Cu concentrations in leaves, while higher concentrations of IAA (50 mg L^−1^) reduced the Cu concentrations in both roots and leaves to the lowest levels. The findings indicated that the application of IAA reduced Cu accumulation, alleviated Cu toxicity, and enhanced Cu tolerance in spinach seedlings. IAA application could be used as an alternative strategy for reducing Cu accumulation in vegetable crops and for remediating Cu-contaminated soil, in turn reducing the hazardous effects of heavy metal contamination on human health and the environment.

## 1. Introduction

The contamination of soil with heavy metals is a global problem. In China, 100 million ha of land have been affected by heavy metal pollution [1] Notably, although copper (Cu) is essential for plant growth and development, the element is also highly phytotoxic at high concentrations, since the free form of Cu is very active in the presence of mercaptan and oxygen, which would promote oxidative stress [2,3]. In recent years, the Cu contents in soil have dramatically increased due to anthropogenic activities, such as Cu mining, smelting waste discharge, the overapplication of Cu-containing pesticides, and the arbitrary discharge of domestic sewage [4,5]. Such excessive levels of Cu contamination pose potential threats to the environment and human health, owing to the element’s high toxicity, rapid accumulation in plant tissues, resistance to degradation, and ability to enter the human body through the consumption of contaminated foods [4].

In plants, excessive Cu levels can facilitate the formation of reactive oxygen species (ROS), which can cause cell oxidative damage and affect a wide range of biochemical and physiological processes, such as antioxidant activity, photosynthesis, mineral uptake, membrane integrity, and nitrogen (N) and protein metabolism [6]. Therefore, numerous studies have explored mechanisms of counteracting the adverse effects of Cu-stress and to improve Cu tolerance of plants via exogenous application of phytohormones [7]. Indole-3-acetic acid (IAA), which is an important signaling molecule in higher plants, functions not only as a plant growth regulator but also as an essential stress tolerance substance [8]. In addition, it alleviates Cu-stress damage [9]. For example, the application of 50 μmol L^−1^ IAA could alleviate the harmful effects of Cu toxicity and increase Cu resistance in wheat plants, while Ouzounidou and Ilias [10] reported that the application of 100 µmol L^−1^ IAA minimized the toxic effects of Cu in the roots of sunflower (*Helianthus annuus* L.), in turn promoting root growth and root hair formation.

The exogenous application of IAA has also been reported to alleviate the adverse effects of heavy metal stress on N metabolism [11]. N is the principal component of all amino acids, proteins, and some nitrogenous compounds that protect plants from abiotic stress [12]. Inorganic forms of N, such as nitrate (NO_3_^−^), are readily taken up by plants, converted into ammonium (NH_4_^+^), and eventually integrated into amino acids and proteins [13], and such processes involve several enzymes, including nitrate reductase (NR), nitrite reductase (NiR), glutamine synthetase (GS), glutamine 2-oxoglutarate aminotransferase (GOGAT), and glutamate dehydrogenase (GDH) [12]. Gangwar and Singh [11] reported that the application of 10 μmol L^−1^ IAA increased the GS, GOGAT, and GDH activities, in addition to N metabolism of Cd-stressed pea seedlings. However, little information is available regarding the influence of IAA on N metabolism under Cu stress.

Spinach (*Spinacia oleracea* L.), which is an important crop rich in vitamins, organic acids, carotenoids, alkaline minerals, and antioxidants [14,15], has been reported to exhibit strong Cu tolerance, owing to its large leaf surface area, relatively high growth rate, and high heavy metal absorption rate [16,17,18]. However, basic research on the biochemical and physiological responses of spinach to Cu is scarce, not to mention the potential role of IAA in spinach seedlings grown under high Cu stress. Therefore, the present study investigated the protective effect of IAA on Cu stress in spinach seedlings. The aim of the present study was to investigate whether the exogenous application of IAA can relieve Cu stress and damage, and enhance Cu tolerance via alteration of the uptake of Cu, change in antioxidant system, and shifts in N metabolism.

## 2. Materials and Methods

### 2.1. Plant Materials

The experiments were conducted in October 2018 in a solar greenhouse at the Faculty of Resources and Environmental Science at Hubei University, China. The test variety was Japanese big-leaf spinach, which was obtained from the seed breeding station of Wangzhendian, Qingxian County, Hebei Province, China. Healthy seeds were surface-sterilized by soaking in 0.1% NaClO for 10 min, washed extensively in deionized water, evenly spaced in Petri dishes (9-cm diameter) that contained filter paper, and then incubated at 25 °C in darkness. After 48 h, germinated seeds were selected and planted in peat soil. After reaching 10 cm in height, uniform healthy seedlings were transplanted into pots and irrigated with complete Hoagland nutrient solution. Each pot contained 10 seedlings, and each pot contained 3 kg (2.1 kg DW) of a (7:3, *v*/*v*) mixture of peat and perlite. The plants were allowed to acclimate for 20 d before the experimental treatments were applied.

### 2.2. Test Treatment

The potted plants were divided into eight groups. One group in which only the Hoagland nutrient solution was added, including 4.56 mg kg^−1^ Cu, was used as the control (C1), and the other seven groups were irrigated with Hoagland nutrient solution containing 700 mg kg^−1^ Cu. The total amount of irrigated solution in each pot in the eight groups was 400 mL. The Hoagland nutrient solution was adjusted to a pH of 6.5 ± 0.3 using 0.1 mmol L^−1^ NaOH or HCl. Cu was supplied as CuSO_4_ (analytical reagent). 

Subsequently, the leaves of the other seven groups were sprayed with different concentrations of IAA (0, 10, 20, 30, 40, 50, and 60 mg L^−1^) and the treatments were designated as C2, T1, T2, T3, T4, T5, and T6, respectively. The IAA solution was mixed with Tween-20 (C_58_H_114_O_26_, leaf surfactant), and the total amount of IAA in each pot was 20 mL. The C2 plants were sprayed with an equal amount of water (20 mL/pot) mixed with Tween-20. Each treatment has three replicates in a randomized block design.

Evaporative water loss was compensated for by adding deionized water until reaching a weight of 3.4 ± 0.05 kg, and the seedlings were cultivated under a 14 h photoperiod (light intensity of 8000 Lux), with a relative humidity of 70–80% and a day/night temperature of 22/15 °C. After 7 d of treatment, plants were collected for the measurement of indicators. 

### 2.3. Measurement Indicators and Methods 

#### 2.3.1. Determination of Biomass

The seedlings were carefully removed from the pots, rinsed using tap water, and then rinsed thrice using deionized water. After drying with blotting paper, the fresh weight (FW) of leaves, stems, and roots were measured immediately, and dry weight (DW) was measured after the samples were fixed in an oven at 105 °C for 10 min, and then dried to a constant weight at 80 °C.

#### 2.3.2. Cu Content Measurements

After separating the leaves and roots of the seedlings, the roots were soaked in 20 mmol L^−1^ Na_2_-EDTA for 3 h to remove surface-adsorbed Cu and then washed repeatedly using deionized water. The root and leaf samples were oven-dried (105 °C for 20 min and 80 °C for 72 h), ground, and then digested using a 4:1 (*v*:*v*) mixture of HNO_3_ and HClO_4_. Finally, the Cu concentrations were determined using inductively coupled plasma atomic emission spectroscopy (ICP-AES; Fisons ARL Accuris, Ecublens, Switzerland) [19]. 

#### 2.3.3. Biochemical Indicators Measurements

Malondialdehyde (MDA) was measured using the thiobarbituric acid method [20], whereas proline content was determined by acidic ninhydrin colorimetry [20]. Superoxide dismutase (SOD, EC 1.15.1.1), peroxidase (POD, EC 1.11.1.7), and ascorbate peroxidase (APX, EC 1.11.1.11) activity were assayed according to the methods described by Beauchamp and Fridovich [21], Batish [22], Nakano, and Asada [23], respectively. The detailed methods of all the biochemical methods are described in our previous study [17]. Lastly, the glutathione reductase (GR, EC 1.6.4.2) activity was measured by monitoring the glutathione dependent oxidation of NADPH. The reaction mixture contained 0.1 mL extract solution, 0.1 mL 2.4 mmol L^−1^ NADPH, 1.7 mL of HEPES, and 0.1 mL of 10 mmol L^−1^ GSSG, and the absorbance of the mixtures was monitored at 340 nm [24].

#### 2.3.4. Leaf N, NO_3_^−^ and NH_4_^+^ Contents Measurement

Total N was quantified using the micro-Kjeldahl method [25] while the NO_3_^−^ and NH_4_^+^ concentrations were determined using an enzyme-linked immunoassay (ELISA) kit (Shanghai Best Choice Biotechnology Co., Ltd., Shanghai, China).

#### 2.3.5. Amino Acids and Protein Concentration Measurements

To measure amino acid concentrations, fresh leaves (0.5 g) were homogenized in 5 mL of 10% acetic acid, diluted to 100 mL with redistilled water, and then filtered. Filtered homogenate solution (1 mL) was then mixed with 3 mL of ninhydrin solution, 1 mL of ammonia-free distilled water, and 0.1 mL of ascorbic acid in a test tube. The mixed solution was heated in a water bath at 100 °C for 15 min. After cooling, the solution was diluted to 20 mL with 60% ethanol, and the absorbance of the solution was measured at 570 nm [26].

Soluble protein was measured according to the method of Bradford [27]. Frozen leaves (0.5 g) were homogenized in 5 mL of 0.1 mol L^−1^ phosphate buffer solution (pH 7.0) and centrifuged at 4000 and 4 °C for 10 min. Sample extract (0.1 mL) was mixed with 5 mL of Coomassie brilliant blue solution (0.1 g Coomassie brilliant blue [G-250] in 50 mL of 90% ethanol and 85% phosphoric acid) and the absorbance values recorded at 595 nm. Protein concentration was determined by comparing the absorbance values to the values of a standard curve established using bovine serum albumin.

#### 2.3.6. N Metabolizing Enzymes Activity Measurement

The activities of N-metabolizing enzymes (NR, NiR, GS, GOGAT, and GDH) were measured using commercially available chemical kits (Shanghai Best Choice Biotechnology) according to the manufacturer’s instructions. Leaf samples (0.1 g) were crushed in a chilled mortar and pestle and extracted in 1 mL buffer (provided by each corresponding kit) in a 2-mL centrifuge tube. The sample was centrifuged and the supernatant was stored in a new tube. The reaction tube contained the recommended reagents for each test kit and conditions were maintained strictly according to the provided instructions. The absorption readings for the five enzymes were recorded at 450 nm using a spectrophotometer. 

### 2.4. Statistical Analysis

Data were reported as mean ± standard deviation (SD) values of three replicate experiments and were analyzed using SPSS Statistics 17.0 (SPSS Inc., Chicago, IL, USA). One-way analysis of variance and Duncan’s multiple range tests were used to determine the significance of differences among the treatment groups, with a significance level of 0.05. The figures were illustrated using Origin Pro 9.0 (OriginLab Co., Northampton, MA, USA).

## 3. Results

### 3.1. Effect of IAA Treatment on Total Biomass Accumulation

The total FW and DW of the C2 treatment were 31.94% and 40.93% higher, respectively, than those of the treatment C1 (*p* < 0.05 Figure 1), and both measures (total FW and DW) increased with an increase in IAA concentrations (T1–T6). Total FW and DW were the greatest in the treatment T6 and were 49.17% and 69.99% greater than in the C2 treatment, respectively. Similar trends were also observed for the biomass of leaves, stems, and roots. 

### 3.2. Effect of IAA Treatment on Cu Accumulation in the Leaves and Roots 

The Cu concentrations of the C2 leaves and roots were 144.79 and 23.36 times greater than those of the C1 leaves and roots, respectively (Table 1). Compared with the C2 treatment, the Cu accumulation of spinach leaves decreased gradually along the treatments T1–T6, with the T6 leaves accumulating 39.61 times less Cu than the C2 leaves. However, a different trend was observed for the Cu concentrations in roots. Instead of decreasing continuously, the Cu concentrations in the T1, T2, T3, and T4 roots increased by 0.20, 2.10, 2.17, and 1.92 times, respectively, and then that of T5 and T6 decreased by 0.29 and 0.87 times, respectively.

### 3.3. Effect of IAA Treatment on MDA and Proline Concentrations 

The MDA concentrations of the C2 treatment was markedly greater than that of the C1 treatment (Figure 2). However, the MDA concentration decreased gradually with an increase in IAA concentrations (T1–T6), with the T5 and T6 treatments yielding 42.58% and 43.85% less MDA, respectively, than the treatment C2. In contrast, the proline contents increased with an increase in IAA concentrations (T1–T6), with the T6 treatment yielding 62.2% more proline than the C2 treatment.

### 3.4. Effect of IAA Treatment on Antioxidant Enzyme Activities

As shown in Figure 3, the SOD, POD, and APX activities (Figure 3) of the C2 treatment were greater (46.70%, 57.07%, and 99.97%, respectively) than those of the C1 treatment (*p* < 0.05), while the GR activity of C2 treatment was 45.91% lower than that of the C1 treatment. Furthermore, the SOD activities of the T1–T6 treatments were consistently higher than those of the C2 treatment, with the T6 treatments exhibiting 44.05% more SOD activity than the C2 treatment. In addition, the POD activities of T1–T6 treatments were lower than those of the C2 treatment, even though the differences were generally insignificant, and the APX activities of the T1–T5 treatments were lower than those of the C2 treatment, while the APX activities of the T6 treatment were remarkably greater (55.56%). Finally, the GR activities remained consistently low (T1–T3), until they increased at higher IAA concentrations (T4–T6). 

### 3.5. Effect of IAA Treatment on Total N Content and N Assimilation Compounds

The total N, NO_3_^−^, free amino acid, and soluble protein contents (Table 2) of the C2 treatment were 3.24%, 43.55%, 48.68%, and 36.08% lower, respectively, than those of the C1 treatment, whereas the NH_4_^+^ content was 1.43 times greater. Apparently, when IAA was applied to Cu-stressed seedlings (T1–T6), total N concentration decreased in the T1–T4 treatments, and were significantly lower than in the C2 treatment (*p* < 0.05), while total N contents in the T5–T6 treatments were not significantly different (*p* > 0.05), from the total N concentrations in the C2 treatment. In addition, the concentrations of NO_3_^−^ and soluble protein increased notably with an increase in IAA concentrations (T1–T6), with the T6 treatment accumulating 1.06 times the NO_3_^−^ in the C2 treatment, while the soluble protein contents were not significantly different between the T6 treatment and the C2 treatment. The NH_4_^+^ concentrations decreased with an increase in IAA concentrations (T1–T6), with the T5 and T6 plants accumulating 47.06% and 41.18% less NH_4_^+^ than the C2 treatment, respectively. Notably, the free amino acid contents of the IAA together with the Cu-stressed seedlings (except for the T5 treatment) were higher than those of the C2 treatment. 

### 3.6. Effect of IAA Treatment on N Metabolizing Enzyme Activities

The NR, NiR, GS, and GOGAT activities of the C2 treatment were 57.78%, 30.64%, 22.92%, and 12.36% lower, respectively, than those of the C1 treatment, whereas GDH activity was 38.64% higher (Table 3). More specifically, the NR activities of T1–T6 treatments exhibited an increasing trend at two stages (T2 and T6), with the greatest increase (1.02 times) observed in the T6 treatment, when compared to the C2 treatment. The NiR and GDH activities of the IAA-treated (except for the T1 treatment) were greater than those of the C2 treatment. However, the activities of both GS and GOGAT were decreased initially and subsequently increased in the T1–T6 treatments. Greater increases in GS and GOGAT activities were observed in the T4–T6 and the T3–T6 treatments, respectively.

## 4. Discussion

The present study was undertaken to identify the potential mechanisms and the influence of IAA in protecting spinach seedlings from Cu toxicity and improving Cu tolerance. Notably, exogenous addition of different concentrations of IAA not only enhanced Cu tolerance of spinach seedlings but also alleviated the damage of spinach seedlings due to Cu stress. The results demonstrated that low concentrations of IAA (10–40 mg L^−1^) increased the Cu concentrations in Cu-stressed seedling roots, but reduced the Cu concentrations of Cu-stressed seedling leaves. The higher Cu levels in the roots of the IAA-treated plants could have occurred because (i) IAA stimulated lateral root formation, which supports more absorption of minerals [28]; (ii) roots act as a barrier against metal translocation to the upper parts of plants as a tolerance strategy [29]; and (iii) exogenous IAA may improve the redistribution of metals to aerial parts of plants, as a potential defensive mechanism against root damage [30]. In addition, the highest concentration of IAA (60 mg L^−1^), which yielded the most substantial reductions in Cu concentrations in the leaves and roots of Cu-stressed seedlings, effectively mitigated the toxic effects of Cu on seedlings. Other studies have also reported that IAA treatment (10 μmol L^−1^) can alleviate the Mn toxicity symptoms and promote the growth of pea seedlings [9]. In addition, soaking wheat grains in 50 μmol L^−1^ IAA reduced Cu accumulation and alleviated Cu toxicity under Cu-stress conditions [8].

Recently, several investigators have reported that exogenously applied IAA at concentrations between 0.01 and 10 μmol L^−1^ can promote growth as well as alleviate toxic effects of abiotic stress in different plant species [28,31,32], while higher IAA concentrations inhibit growth and cause oxidative damage in plants [33]. In contrast, in the present study, even at higher IAA concentrations, the biomass of Cu-treated seedlings continued to increase with an increase in IAA concentrations, and was the greatest at the highest tested concentration (60 mg L^−1^ IAA). In addition, increasing IAA concentrations may have been required to induce plants to enhance their capacity to resist high Cu-stress [34,35]. 

Plant cells have evolved complex antioxidant systems (enzymatic and non-enzymatic antioxidants) to cope with ROS and alleviate the deleterious effects of Cu. Antioxidant enzymes include SOD, POD, ctalase (CAT), APX, and GR [36], and non-enzymatic antioxidants are a variety of compounds, such as proline, glutathione, ascorbic acid, and amino acids, which regulate cellular concentrations of O_2_^-^ and H_2_O_2_, thereby ameliorating the harmful effects of ROS [37]. In the present study, the application of IAA to Cu-stressed seedlings increased the activities of antioxidant enzymes (SOD, APX, and GR) and reduced MDA concentrations, indicating that exogenous addition of IAA could induce the synthesis of antioxidant enzymes, or activate specific gene-mediated antioxidant enzyme activities [38], and, in turn, enhanced the antioxidant defense ability and reduced the lipid peroxidation damage in the seedlings [39]. Such defense capacity was more obvious, particularly at the highest IAA concentrations (60 mg L^−1^). Indeed, as non-enzymatic antioxidant, the proline contents of IAA-treated seedlings were remarkably higher, which suggested that IAA application stimulated proline accumulation [40], and alleviated free-radical damage induced by Cu stress [8]. Therefore, the findings suggest that exogenous application of IAA improved Cu tolerance in seedlings by enhancing antioxidant defense systems and alleviating the damage caused by Cu stress.

Excessive Cu has been reported to interfere with N acquisition, assimilation, and the activities of N-metabolizing enzymes [41,42]. The reduction of NO_3_^−^ to NH_4_^+^ involves NR and NiR, while NH_4_^+^ assimilation usually involves the GS-GOGAT cycle or the GDH alternative pathway [43,44]. In the present study, the total N and NO_3_^−^ concentrations were reduced by Cu exposure (C2 treatment), and the activities of NR and NiR were also reduced, potentially due to the inhibition of inorganic N absorption by the Cu stressed seedlings [41], and the reduction capacity of NO_3_^−^ declined with decrease in Cu stress [45,46]. However, NH_4_^+^ content under Cu stress was observed to increase, compared with the C1 treatment, potentially because (i) the N metabolism of seedlings was vulnerable to Cu stress, which induced the accumulation of dangerously high levels of NH_4_^+^ [47] (ii) NH_4_^+^ continuously formed during various metabolic processes, such as direct absorption, photorespiration, soluble organic N catabolism and N compounds storage [48]. The present study also revealed reductions in GS and GOGAT activities, which indicated that the GS/GOGAT cycle was inhibited, thereby disrupting NH_4_^+^ assimilation, and causing the observed increases in NH_4_^+^ concentrations [39,49]. Notably, NH_4_^+^ can also be rapidly assimilated into organic N through the GDH alternative pathway [50]. However, in the present study, even though GDH activity was enhanced in the C2 treatment, the NH_4_^+^ concentrations remained relatively high, potentially because the GDH pathway was limited in its capacity to assimilate NH_4_^+^ and only assimilated a small amount of NH_4_^+^ [45]. Indeed, Cd toxicity has also been reported to inhibit the GS/GOGAT cycle, increase GDH activity, and increase NH_4_^+^ concentrations in the leaves of *Solanum nigrum* L. [45]. In the present study, the accumulation of NH_4_^+^ was also accompanied by a remarkable reduction in free amino acid and soluble protein contents, which indicated that the Cu stress inhibited N assimilation of seedlings and reduced the biosynthesis of free amino acids and soluble proteins [45,51].

Furthermore, the exogenous application of IAA can improve plant tolerance and, in particular, can counteract the negative effects of heavy metals on plant growth [9]. However, it is unclear how IAA application affects N metabolism. Therefore, one of the aims of the present study was to measure the effects of IAA on N assimilation metabolism. Exogenous IAA application increased the contents of total N and NO_3_^−^, and the activities of NR and NiR, which were involved in NO_3_^−^ reduction and N metabolism [11,52], thereby indicating that IAA application could enhance the NO_3_^−^ absorption efficiency of Cu-stress seedlings, improve the activities of NR and NiR, and enhance the N reduction capacity [11]. Similar results have been reported by previous studies. For example, the addition of 10 μmol L^−1^ IAA to Cr stressed pea seedlings increased the total N contents of roots and shoots [11]. Moreover, the synchronous increases observed in NO_3_^−^ content and NR activity in spinach seedlings could be due to the compartmentalization of NO_3_^−^ assimilation in leaves [53], since NO_3_^−^ is mainly stored in vacuoles [54], and isolated from NR, which is located in the cytoplasm [55]. 

The avoidance of excess NH_4_^+^ accumulation is generally viewed as an important strategy for withstanding stress damage in plants [56]. In the present study, the application of IAA treatment increased the GS and GOGAT activities of Cu-stressed seedlings, but reduced NH_4_^+^ content, which implied that the GS-GOGAT enzyme system plays an essential role in NH_4_^+^ detoxification [56,57]. Under stressful conditions, NH_4_^+^ is assimilated through an alternative pathway that involves GDH [45]. In the present study, the GDH activities of the Cu-stressed plants were elevated by IAA application, which indicated that increases in GDH activity may alleviate the adverse effects of accumulating toxic amounts of NH_4_^+^ and may supply glutamate for the biosynthesis of protective compounds [58]. In addition, according to the results of the present study, free amino-acid and soluble protein contents increased with an increase in IAA concentrations, which may be due to IAA-induced increase in the activities of N-metabolizing enzymes and in the efficiency of N assimilation [59], or due to the mitigating effects of exogenous IAA on Cu stress, as reported by Agami [8].

## 5. Conclusions

The present study demonstrated that the application of lower concentrations of IAA (10–40 mg L^−1^) could increase the Cu concentrations in Cu-stressed seedling roots and reduce the Cu concentrations of Cu-stressed seedling leaves, while higher concentrations of IAA (50 mg L^−1^) could notably reduce the Cu concentration in both roots and leaves. Together, the findings indicate that exogenous IAA application can be used to reduce Cu accumulation, alleviate Cu toxicity, and enhance Cu tolerance in spinach seedlings and that such benefits are the results of the enhancement of antioxidant defense systems, reducing lipid peroxidation damage, and improving the efficiency of N metabolism. This demonstrates that exogenous IAA application can be an alternative strategy for reducing Cu accumulation in vegetable crops and for remediating Cu-contaminated soil, reducing the hazardous effects of heavy metal contamination on human health and the environment.

## Figures and Tables

**Figure 1 toxics-08-00001-f001:**
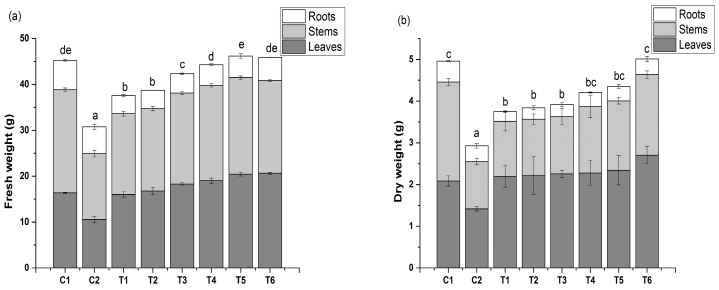
Effects of indole-3-acetic acid (IAA) on the fresh weight (**a**) and dry weight (**b**) of spinach seedlings under Cu stress. Values and error bars indicate means ± standard deviation (*n* = 3). Different letters indicate a significant difference between treatments at *p* < 0.05 according to Duncan’s multiple-comparisons test.

**Figure 2 toxics-08-00001-f002:**
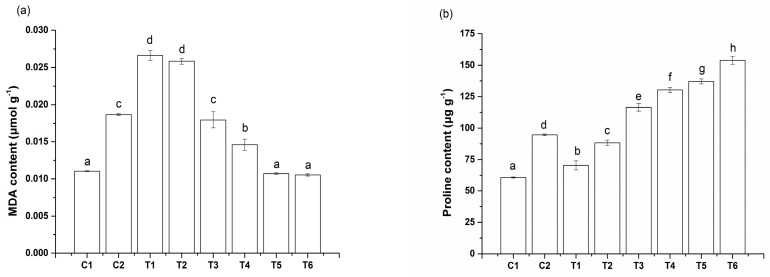
Effects of IAA on malondialdehyde (MDA) (**a**) and proline (**b**) contents of spinach seedlings under Cu stress. Values and error bars indicate means ± standard deviation (*n* = 3). Different letters indicate a significant difference between treatments at *p* < 0.05 according to Duncan’s multiple-comparisons test.

**Figure 3 toxics-08-00001-f003:**
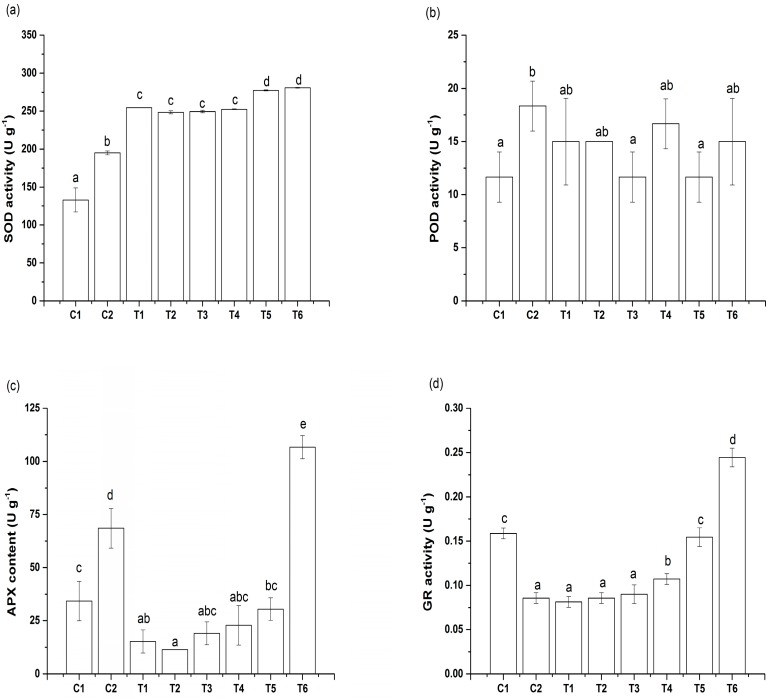
Effects of IAA on superoxide dismutase (SOD; **a**), Peroxidase (POD; **b**), Ascorbate peroxidase (APX; **c**) and glutathione reductase (GR; **d**) activities of spinach seedlings under Cu stress. Values and error bars indicate means ± standard deviation (*n* = 3). Different letters indicate a significant difference between treatments at *p* < 0.05 according to Duncan’s multiple-comparisons test.

**Table 1 toxics-08-00001-t001:** Effects of IAA on the Cu Accumulation by Cu-Stressed Spinach Seedlings.

Treatment	Leaf Concentration	Root Concentration
	**(μg g^−1^)**	**(μg g^−1^)**
C1	19.72 ± 0.14 a	7.97 ± 0.06 a
C2	2875.00 ± 2.16 h	194.15 ± 1.06 d
T1	1777.83 ± 15.46 g	232.02 ± 0.78 e
T2	1616.50 ± 13.08 f	601.50 ± 8.98 g
T3	1475.50 ± 16.57 e	616.17 ± 2.49 h
T4	1319.17 ± 1.25 d	566.33 ± 9.80 f
T5	122.03 ± 2.62 c	137.55 ± 4.71 c
T6	72.58 ± 5.03 b	25.87 ± 0.06 b

Data are mean ± standard error of three replicates. Values within a row followed by the same letter are not significantly different (*p* < 0.05) based on Duncan’s multiple-comparisons test.

**Table 2 toxics-08-00001-t002:** Effects of IAA on total N content and N assimilation compounds under Cu stress.

Treatment	Total N Content (g kg^−1^)	NO_3_^−^ Content (g kg^−1^)	NH_4_^+^ Content (g kg^−1^)	Free Amino Acid Content (g kg^−1^)	Soluble Protein Content (g kg^−1^)
C1	83.24 ± 0.91 e	1.24 ± 0.016 f	0.07 ± 0.0018 a	0.76 ± 0.010 f	18.82 ± 0.60 cd
C2	80.54 ± 0.66 cde	0.70 ± 0.039 a	0.17 ± 0.0001 h	0.39 ± 0.004 b	12.03 ± 0.14 a
T1	77.47 ± 1.92 ab	0.75 ± 0.006 b	0.16 ± 0.0002 g	0.51 ± 0.014 d	14.83 ± 1.18 b
T2	78.06 ± 1.64 abc	1.10 ± 0.018 d	0.12 ± 0.0003 f	0.49 ± 0.012 d	13.72 ± 1.18 ab
T3	76.02 ± 1.48 a	1.01 ± 0.0041 c	0.11 ± 0.0004 d	0.41 ± 0.020 b	13.74 ± 0.70 ab
T4	78.91 ± 1.58 bcd	1.19 ± 0.0039 e	0.12 ± 0.0006 e	0.67 ± 0.016 e	14.88 ± 1.28 b
T5	81.14 ± 0.35 de	1.23 ± 0.0084 ef	0.09 ± 0.0006 b	0.30 ± 0.011 a	17.70 ± 0.05 c
T6	81.24 ± 0.34 de	1.44 ± 0.039 g	0.10 ± 0.0005 c	0.44 ± 0.014 c	20.51 ± 0.77 d

N, nitrogen; NO_3_^−^, nitrate nitrogen; NH_4_^+^, ammonium nitrogen. Data are mean ± standard error of three replicates. Values within a row followed by the same letter are not significantly different (*p* < 0.05) by Duncan’s multiple-comparisons test.

**Table 3 toxics-08-00001-t003:** Effects of IAA on N Metabolizing Enzymes Activities under Copper Stress.

Treatment	NR (U g^−1^)	NiR (U g^−1^)	GS (U g^−1^)	GOGAT (U g^−1^)	GDH (U g^−1^)
C1	1224.75 ± 18.71 e	2537.13 ± 15.71 f	0.48 ± 0.004 f	1.78 ± 0.028 d	0.044 ± 0.0003 a
C2	517.03 ± 14.06 a	1759.71 ± 3.93 b	0.37 ± 0.003 c	1.56 ± 0.019 c	0.061 ± 0.0014 c
T1	659.59 ± 8.2 5b	1582.01 ± 61.21 a	0.23 ± 0.006 a	1.30 ± 0.026 b	0.051 ± 0.0009 b
T2	803.43 ± 34.34 c	1848.56 ± 37.87 c	0.29 ± 0.006 b	1.24 ± 0.016 a	0.072 ± 0.0010 d
T3	658.32 ± 10.95 b	2128.98 ± 21.86 d	0.37 ± 0.005 c	1.81 ± 0.016 d	0.088 ± 0.0003 f
T4	781.79 ± 12.47 c	2495.48 ± 48.25 ef	0.41 ± 0.007 d	2.09 ± 0.013 g	0.088 ± 0.0013 f
T5	819.98 ± 8.25 c	2437.17 ± 34.23 e	0.41 ± 0.004 d	1.99 ± 0.006 f	0.079 ± 0.0019 e
T6	1045.28 ± 22.48 d	2792.57 ± 23.89 g	0.42 ± 0.003 e	1.92 ± 0.005 e	0.072 ± 0.0015 d

NR, nitrate reductase; NiR, nitrite reductase; GS, glutamine synthetase; GOGAT, glutamine 2-oxoglutarate aminotransferase; GDH, glutamate dehydrogenase. Data are mean ± standard error of three replicates. Values within a row followed by the same letter are not significantly different (*p* < 0.05) by Duncan’s multiple-comparisons test.

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
