# Peer review of "Exogenous of Indole-3-Acetic Acid Application Alleviates Copper Toxicity in Spinach Seedlings by Enhancing Antioxidant Systems and Nitrogen Metabolism"

_toxics, 2019, doi:10.3390/toxics8010001_

Round 1
Reviewer 1 Report
This is well written manuscript regarding very important and interesting issue. Body of the manuscript is constructed property according to the journals guidelines. Experiment was carefully planned with sufficient level of descriptive detail included in the manuscript. Methodology used in the study was also well fitted thus conclusion drawn in the study are verifiable and correct.
Manuscript does not require revision however despite the fact that Introduction section is pretty informative I will recommend small addition, literally two or three sentences on the mechanism of uptake of copper from soil since toxicity of this element for plants depends also on the coexistence of other antagonistic elements such as Zn, Fe or Cd. Did authors monitor also concentration of these elements during the experiment? That is only a question that of course does not undermines the research.
Line 67 please do not start the sentence with “So..” just remove this conjunction.
Author Response
Thank you for your comments and suggestions on our manuscript. We have revised the manuscript accordingly, and detailed corrections are listed below:
Point 1: I will recommend small addition, literally two or three sentences on the mechanism of uptake of copper from soil since toxicity of this element for plants depends also on the coexistence of other antagonistic elements such as Zn, Fe or Cd. Did authors monitor also concentration of these elements during the experiment?
Response 1: Thank you for your kind comments and reminding. Indeed, the toxicity of excessive Cu in soil to plants also depends on the coexistence of other antagonistic elements such as Zn, Fe and Cd. Therefore, in our next research, we will monitor the changes of other mineral elements in soil and plants and the analysis of the interaction of mineral elements.
Point 2: Line 67 please do not start the sentence with “So..” just remove this conjunction.
Response 2: We agree with this suggestion, and have revised in the manuscript (Line 69).
Reviewer 2 Report
Generally, the manuscript is well-written. Authors main idea was to research influence of IAA on Cu-stressed spinach plants. I propose to improve the aim of the manuscript, some parts of research and discussion. There are some editorial mistakes that should be corrected throughout the text (e.g. space bar application).
Introduction
Please, rewrite the aim of the study in order to be more precise, esp. in (i). Based on the presented research, rethink if the division into (i) and (ii) is a good idea.
Materials and methods
Lines 117-119 – Shorten the text into one sentence, for example “Activities of peroxidase (POD, EC 1.11.1.7) and ascorbate peroxidase (APX, EC 1.11.1.11) were assayed according to Batish[23] and Nakano and 118 Asada [24], respectively”.
Line 146 – Please, correct the word “nextracted”.
Results
Please, check the calculations presented in the text, e.g. I think that in line 171 it should be 39.61.
Lines 180-181 – Based on the statistical test – not only T6, but also T5, check, please.
Lines 205-207 – Please, check the sentence – double C2 is a mistake in the sentence.
Lines 207-209 and 209-212 – Please, add the exceptions. Take under consideration results which are statistically significant.
Lines 224-226 – Please, check the sentence and correct it. The trend for GS is T4-T6 and for GOGAT is T3-T6.
Discussion
Line 242 - Correct language in the expression "Similar this finding ...".
Line 243 - Lack of the full stop before "Soaking".
Line 249 - Correct language in the expression "... at much high ...". I think that you have meant "... higher ...".
Lines 249-251 - Please, change the sentence precisely. Is it true only for the highest concentration? See Fig. 1a - T5 is statistically the same as T6, but in Fig. 1b - T6 is correct.
Line 250 - space bar before "treated" should be deleted.
Line 255 - Omit the repetition of the word "include".
Lines 257-262 - Please, specify the sentence. The action of IAA depends on its concentration. You should underline it in the sentence.
Line 264 - "remarkable" or "remarkably"?
Line 271 - Instead of "dramatically" you could write "were".
Line 274-277 - Please, check and specify the statement "... NH4+ content was observed to increase ...".
Line 276 - Please, delete full stop before "(ii)".
Line 284 - Please, delete the space bar before "increase".
Lines 291 and 292 - Omit the repetition of the expression "the present study".
Line 305 - "alternate path-way" or "alternative pathway"?
Conclusions
Line 316 - Is it true only for 60 mg L-1, but not for 50 mg L-1?
References
References certainly need better editing.
lack of space bar, e.g. lines 344, 352, 383, 385, 407, 435, 438; unnecessary space bar, e.g. line 356; way of title of journal citations need unification. See e.g. line 448 vs. 455; add italics for latin names. See e.g. line 453; "-" or ":" before the number of pages. See, e.g. line 461 vs. 412 or 463. unify comas and full stops near names and surnames. See, e.g. lines 470, 473, 476 - three different ways of citations.
Abstract
Please, change abstract according to any other changes made in the manuscript.
Author Response
Thank you for your comments and suggestions on our manuscript. We have revised the manuscript accordingly, and detailed corrections are listed below:
Point 1: Please, rewrite the aim of the study in order to be more precise, esp. in (i). Based on the presented research, rethink if the division into (i) and (ii) is a good idea.
Response 1: The aim of the study has been rewritten as suggested (Line 70-71).
Point 2: Line 67 please do not start the sentence with “So..” just remove this conjunction.
Response 2: We have revised (Line 69-71).
Point 3: Lines 117-119 – Shorten the text into one sentence, for example “Activities of peroxidase (POD, EC 1.11.1.7) and ascorbate peroxidase (APX, EC 1.11.1.11) were assayed according to Batish[23] and Nakano and 118 Asada [24], respectively”.
Line 146 – Please, correct the word “nextracted”.
Response 3: The sentence has been improved as suggested (Line 119-121). The word “nextracted” has been revised “extracted” (Line 151).
Point 4: Results
Please, check the calculations presented in the text, e.g. I think that in line 171 it should be 39.61.
Lines 180-181 – Based on the statistical test – not only T6, but also T5, check, please.
Lines 205-207 – Please, check the sentence – double C2 is a mistake in the sentence.
Lines 207-209 and 209-212 – Please, add the exceptions. Take under consideration results which are statistically significant.
Lines 224-226 – Please, check the sentence and correct it. The trend for GS is T4-T6 and for GOGAT is T3-T6.
Response 4: The contents in this manuscript have been revised as suggested.
Line 171 ⇒ Line 178.
Lines 180-181 ⇒ Line 189.
Lines 205-207 ⇒ Line 217.
Lines 207-209 ⇒ Lines 218-221.
Lines 209-212 ⇒ Lines 223-226.
Lines 224-226 ⇒ Lines 244-245.
Point 5: Discussion
Line 242 - Correct language in the expression "Similar this finding ...".
Line 243 - Lack of the full stop before "Soaking".
Line 249 - Correct language in the expression "... at much high ...". I think that you have meant "... higher ...".
Response 5: The contents in this manuscript have been revised as suggested.
Line 242 ⇒ Lines 265-266.
Line 243 ⇒ Line 267.
Line 249 ⇒ Lines 272-273.
Point 6: Discussion
Lines 249-251 - Please, change the sentence precisely. Is it true only for the highest concentration? See Fig. 1a - T5 is statistically the same as T6, but in Fig. 1b - T6 is correct.
Response 6: The sentence has been improved as suggested (Lines 272-273).
The meaning of this sentence is that the biomass of spinach seedlings under Cu stress can increase with the increase of IAA concentration even if higher concentrations of IAA were added. The biomass of spinach reached the maximum value when treated with the highest IAA concentration. It is not only at the highest IAA concentration that seedling biomass increased.
Point 7: Discussion
Line 250 - space bar before "treated" should be deleted.
Line 255 - Omit the repetition of the word "include".
Response 7: The contents in this manuscript have been revised as suggested.
Line 250 ⇒ Line 273.
Line 250 ⇒ Line 279.
Point 8: Discussion
Lines 257-262 - Please, specify the sentence. The action of IAA depends on its concentration. You should underline it in the sentence.
Response 8: Sorry, we had carefully checked the manuscript in Lines 257-262, but we did not find this sentence “The action of IAA depends on its concentration”.
Point 9: Discussion
Line 264 - "remarkable" or "remarkably"?
Line 271 - Instead of "dramatically" you could write "were".
Line 274-277 - Please, check and specify the statement "... NH4+ content was observed to increase ...".
Line 276 - Please, delete full stop before "(ii)".
Line 284 - Please, delete the space bar before "increase".
Lines 291 and 292 - Omit the repetition of the expression "the present study".
Line 305 - "alternate path-way" or "alternative pathway"?
Response 9: The contents in this manuscript have been revised as suggested.
Line 250 ⇒ Line 288.
Line 271 ⇒ Line 296.
Line 274-277 ⇒ Line 299-300. “NH4+ content was observed to increase” revised to “NH4+ content under Cu stress was observed to increasing, compared with the treatment C1”.
Line 276 ⇒ Line 301.
Line 284 ⇒ Line 310.
Lines 291-292 ⇒ Line 318-319.
Line 305 ⇒ Line 332.
Point 10: Conclusions
Line 316 - Is it true only for 60 mg L-1, but not for 50 mg L-1?
Response 10: Sorry for the mistake we made in writing. We have revised. 60 mg L-1 revised to 50 mg L-1.
Point 11:References certainly need better editing.
lack of space bar, e.g. lines 344, 352, 383, 385, 407, 435, 438; unnecessary space bar, e.g. line 356; way of title of journal citations need unification. See e.g. line 448 vs. 455; add italics for latin names. See e.g. line 453; "-" or ":" before the number of pages. See, e.g. line 461 vs. 412 or 463. unify comas and full stops near names and surnames. See, e.g. lines 470, 473, 476 - three different ways of citations.
Response 11: We have revised all the references.
Point 12: Abstract
Please, change abstract according to any other changes made in the manuscript.
Response 12: We have revised the abstract.
Reviewer 3 Report
In this paper, the authors evaluated the effects of increasing concentration of Indole-3-Acetic Acid (IIA) application on Cu-stressed spinach seedlings. Increased biomass of roots, stems and leaves, higher antioxidant enzyme activities, and decreased levels of ammonium ion and malondialdehyde were detected following IIA treatment. The study is interesting and generally technically sound, the introduction contains sufficient element to characterize background and the results are fairly well presented. Nonetheless, language revision by a native speaker is required especially in discussion (i.e., there is an excessive use of “the present study” and a number of grammar/typological errors)
Thereinafter I point out specific comments to the text in order of appearance.
Line 16. Please amend the typo in “malondialdehyde”
Line 45. Consider not starting the sentence with “such as”.
Line 47. Please replace the comma with the dot after “substance”, and then start a new sentence. Line 47 also needs language revision.
Lines 86, 87. Please explain the two different concentration values of Hoagland nutrient solution.
Line 104. “measured”, not “measure”.
Line 114. Consider deleting the dot and combine the two sentences.
Lines 125-126. This statement is confusing.
Figures 2-3. Where MDA proline and the other biochemical indicators were measured? The meaning of the letters in the graphs is unclear, and the significant differences could be indicated by a star. Additionally, a list of abbreviations below the figures is recommended.
Line 206. Please replace C2 with C1.
Lines 207, 214. Consider to add “seedlings” after “Cu-stressed”.
Table 1-3. As requested for Figure 2 and 3, please explain the use of different letters.
Line 237. “supports”, not “support”. Moreover, please add “a” before “barrier”.
Lines 238-240. I do not understand how Cu accumulation in roots may lead to point III. In addition, probably “improve” not “improved”.
Line 242. Consider to replace “Similar this finding” with “Consistently with this finding”.
Line 248. Please add “whereas” before “higher”.
Line 255. Please give full name of CAT on the first use.
Line 259. “indicating” not “indicated”.
Line 260. “enzymes”, not “enzyme”.
Line 271. Please add “were” before “reduced”.
Line 272. Consider to delete “and”.
Author Response
Thank you for your comments and suggestions on our manuscript. We have revised the manuscript accordingly, and detailed corrections are listed below:
Point 1: Line 16. Please amend the typo in “malondialdehyde”
Line 45. Consider not starting the sentence with “such as”.
Line 47. Please replace the comma with the dot after “substance”, and then start a new sentence. Line 47 also needs language revision.
Response 1: The contents in this manuscript have been revised as suggested.
Line 16 ⇒ Line 16
Line 45 ⇒ Line 46
Line 47 ⇒ Lines 48-49
Point 2: Lines 86, 87. Please explain the two different concentration values of Hoagland nutrient solution.
Response 2: The sentence has been revised as suggested. (Lines 87-92).
Point 3: Line 104. “measured”, not “measure”.
Line 114. Consider deleting the dot and combine the two sentences.
Lines 125-126. This statement is confusing.
Response 3: The contents in this manuscript have been revised as suggested.
Line 104 ⇒ Line 107
Line 114 ⇒ Line 117
Lines 125-126 ⇒ Lines 130-131
Point 4: Figures 2-3. Where MDA proline and the other biochemical indicators were measured? The meaning of the letters in the graphs is unclear, and the significant differences could be indicated by a star. Additionally, a list of abbreviations below the figures is recommended.
Response 4: MDA proline and the other biochemical indicators were measured in the laboratory of Faculty of Resources and Environment Science and School of Life Sciences of Hubei University.
Thank you for your kind comments and reminding. We would like to apologize that we did not describe it in detail in figure and table footnote, and Figure and Table footnote sentence have been improved as suggested.
Point 5: Line 206. Please replace C2 with C1.
Lines 207, 214. Consider to add “seedlings” after “Cu-stressed”.
Response 5: We have revised as suggested.
Line 206 ⇒ Line 217.
Lines 207, 214 ⇒ Lines 219, 229.
Point 6: Table 1-3. As requested for Figure 2 and 3, please explain the use of different letters.
Response 6: Figure and Table footnote sentence have been improved as suggested.
Point 7: Line 237. “supports”, not “support”. Moreover, please add “a” before “barrier”.
Response 7: We have revised as suggested (Lines 260).
Point 8: Lines 238-240. I do not understand how Cu accumulation in roots may lead to point III. In addition, probably “improve” not “improved”.
Response 8: Point III--- We want to explain that the exogenous application of IAA can increase the Cu content in the roots, and decrease the Cu content in the leaves of Cu-stressed seedlings, so we speculated that this might be the result of exogenous IAA leading to the redistribution of Cu content in the roots and leaves of plant.
We used “improve” to replace the “improved” (Lines 262).
Point 9: Line 242. Consider to replace “Similar this finding” with “Consistently with this finding”.
Line 248. Please add “whereas” before “higher”.
Line 259. “indicating” not “indicated”.
Line 260. “enzymes”, not “enzyme”.
Line 271. Please add “were” before “reduced”.
Line 272. Consider to delete “and”.
Response 9: We have revised.
Line 242 ⇒ Line 265
Line 248 ⇒ Line 271
Line 259 ⇒ Line 283
Line 260 ⇒ Line 284
Line 271 ⇒ Line 296
Line 272 ⇒ Line 297
Point 10: Line 255. Please give full name of CAT on the first use.
Response 10: Thank you for your kind comments and reminding. We have added full name of CAT in line 279.